# The Neuroprotective Effect of Neural Cell Adhesion Molecule L1 in the Hippocampus of Aged Alzheimer’s Disease Model Mice

**DOI:** 10.3390/biomedicines12081726

**Published:** 2024-08-01

**Authors:** Miljana Aksic, Igor Jakovcevski, Mohammad I. K. Hamad, Vladimir Jakovljevic, Sanja Stankovic, Maja Vulovic

**Affiliations:** 1Center for Medical Biochemistry, University Clinical Center of Serbia, 11000 Belgrade, Serbia; miljanaaksic@gmail.com (M.A.); sanjast2013@gmail.com (S.S.); 2Institut für Anatomie und Klinische Morphologie, Universität Witten/Herdecke, 58455 Witten, Germany; 3Department of Neuroanatomy and Molecular Brain Research, Institute of Anatomy, Ruhr Universität Bochum, 44801 Bochum, Germany; 4Department of Anatomy, College of Medicine and Health Sciences, United Arab Emirates University, Al Ain P.O. Box 64141, United Arab Emirates; 5Center of Excellence for Redox Balance Research, Cardiovascular and Metabolic Disorders, Department of Physiology, Faculty of Medical Sciences, University of Kragujevac, 34000 Kragujevac, Serbia; drvladakgbg@yahoo.com; 6Faculty of Medical Sciences, University of Kragujevac, 34000 Kragujevac, Serbia; 7Department of Anatomy, Faculty of Medical Sciences, University of Kragujevac, 34000 Kragujevac, Serbia; maja@medf.kg.ac.rs

**Keywords:** adhesion molecule L1, Alzheimer’s disease, APP/PS1 mice, GABAergic interneurons, hippocampus, synapses

## Abstract

Alzheimer’s disease (AD) is a severe neurodegenerative disorder and the most common form of dementia, causing the loss of cognitive function. Our previous study has shown, using a doubly mutated mouse model of AD (APP/PS1), that the neural adhesion molecule L1 directly binds amyloid peptides and decreases plaque load and gliosis when injected as an adeno-associated virus construct (AAV-L1) into APP/PS1 mice. In this study, we microinjected AAV-L1, using a Hamilton syringe, directly into the 3-month-old APP/PS1 mouse hippocampus and waited for a year until significant neurodegeneration developed. We stereologically counted the principal neurons and parvalbumin-positive interneurons in the hippocampus, estimated the density of inhibitory synapses around principal cells, and compared the AAV-L1 injection models with control injections of green fluorescent protein (AAV-GFP) and the wild-type hippocampus. Our results show that there is a significant loss of granule cells in the dentate gyrus of the APP/PS1 mice, which was improved by AAV-L1 injection, compared with the AAV-GFP controls (*p* < 0.05). There is also a generalized loss of parvalbumin-positive interneurons in the hippocampus of APP/PS1 mice, which is ameliorated by AAV-L1 injection, compared with the AAV-GFP controls (*p* < 0.05). Additionally, AAV-L1 injection promotes the survival of inhibitory synapses around the principal cells compared with AAV-GFP controls in all three hippocampal subfields (*p* < 0.01). Our results indicate that L1 promotes neuronal survival and protects the synapses in an AD mouse model, which could have therapeutic implications.

## 1. Introduction

Alzheimer’s disease (AD) is a neuropsychiatric disorder with rising incidence and is becoming one of the leading causes of neurological mortality in the elderly. Dominant pathological changes in AD are intracellular neurofibrillary tangles, which consist of tau protein, and extracellular aggregates (otherwise known as plaques) composed of amyloid beta peptide (Aβ). The Aβ is a peptide containing 40 or 42 amino acids cleaved from the amyloid precursor protein (APP). As Aβ-42 is more prone to aggregate than Aβ-40, initial Aβ deposition begins with Aβ-42, so that the initial amyloid plaques are formed only by Aβ-42, whereas the presence of Aβ-40 signifies the advanced stage of the disease [1]. AD is steadily increasing in both incidence and prevalence and is quickly becoming one of the most expensive, lethal, and burdening diseases of this century [2]. It is estimated that the current number of 50 million AD patients worldwide will triple by the year 2050 [2]. Despite our significant understanding of the pathogenesis of AD, the effects of currently available therapies, such as cholinesterase blockers and N-methyl-D-aspartate (NMDA) receptor agonists, are less well-known [3]. Additionally, several amyloid-binding antibodies have recently been approved for patient treatment in the USA (e.g., lecanemab, donanemab, and aducanumab), and clinical trials are underway for other Aβ binding antibodies and immunization protocols, but their clinical success is still to be tested [3]. Therefore, there is still a need for novel therapeutic approaches.

Adhesion molecules are involved in development by facilitating neurite outgrowth, pathfinding, and synaptogenesis, thus having excellent potential for the treatment of neurodegenerative disorders [4]. Our previous work has shown that the neural cell adhesion molecule L1 represents one such molecule. We previously demonstrated, using a mouse model of cerebral amyloidosis (APP/PS1 mouse), that L1 directly binds to the Aβ peptides, ameliorates amyloid plaque load, and decreases gliosis [5]. Furthermore, elevated levels of L1 have been detected in the cerebrospinal fluid of patients with AD and several other dementias, suggesting a functional role of L1 in AD [6]. Additionally, L1 overexpressing embryonic stem cells, transplanted to a mouse model of Huntington’s disease, differentiated preferentially into neurons over glia, with increased differentiation into gamma-aminobutyric acid (GABA)-ergic neurons and behavioral improvement [7]. Importantly, some more recent work suggests that a proteolytically cleaved fragment of the L1 molecule reduces Aβ [8] and that the treatment of amyloid-expressing cells with chicoric acid increases synaptic proteins and L1 expression in these cells [9]. Given these potentially beneficial effects of L1, demonstrated both in vivo and in vitro, we next wanted to investigate its effects in a fully developed neurodegeneration model.

One of the major changes seen in Alzheimer’s disease is the loss of synapses and neurons, which begins in the hippocampus [10,11]. Although there is no global neuron loss in APP/PS1 mice, in aged (17-month-old) as well as in younger mice, the loss of dentate gyrus granule cells was observed [10,11]. Other investigators have reported the loss of GABAergic interneurons, particularly parvalbumin-positive cells in the hippocampus [11]. In our previous study, we demonstrated that L1 transduced mice have better GABAergic synapse preservation in the hippocampus than the controls at the age of 8 months [5]. To demonstrate the therapeutic potential of L1 in Alzheimer’s disease, we used a mouse AD model with the accelerated progression of disease symptoms, which we used in our previous publications, a transgenic mouse APP/PS1 on a C57BL/6J inbred genetic background. These mice co-express the “Swedish mutation” human APP, with L166P mutated human presenilin-1, showing many of the histological aspects of cerebral amyloidosis, including amyloid plaque formation, synaptic degeneration, hyperphosphorylated tau-positive neuron tangles, astrogliosis, and microgliosis [12,13]. As a method for the delivery of L1 to the diseased brain tissue, we used AAV-driven transduction, as the virus spreads well through the brain tissue over time and progressively transduces more cells. Previous experience with AAV-L1 confirmed that serotype 5 of this virus stably transduces neurons and astrocytes [5,14]. Importantly, overall L1 protein expression in APP/PS1 mice does not differ from the control, although the proteolytic processing of L1 may be altered [15].

An obvious question remains: how could such a large glycoprotein as L1 be a suitable therapeutic approach? Currently, there are several feasible application formulations for carrying such large constructs into the brain, e.g., slowly releasing synthetic hydrogels, nanoparticles, organic biomaterials like alginate, or a recombinant construct expressing stem and various other cells [16]. All these methods successfully bypass the blood–brain barrier and, thus, circumvent any potential unknown consequences of L1 effects outside of the nervous system [16]. Although various currently available strategies directed toward the reduction or clearance of amyloid plaque deposition were insufficient to cure fully developed AD, it is conceivable that some future development of early diagnostic methods will increase therapeutic success [17]. In addition, several other adhesion molecules were recently posited as possible therapeutic targets in AD, for example, polysialic acid [18], CD31 [19], and reelin [20].

To extend our previous observation that L1 promotes amyloid peptide clearance in the long term and to further show its neuroprotective effects, we have now injected a new set of 3-month-old APP/PS1 mice with AAV-L1 and let them age for 12 months, before sacrifice at 15 months of age. We injected the mice at 3 months, as it is an adult age at which the mice still do not show any signs of amyloid-related pathology [12]. We stereologically counted all neurons, as well as the parvalbumin-expressing inhibitory interneurons in the hippocampus. Our results indicate that AAV-L1 injection leads to a better survival of the dentate gyrus granule cells and the GABAergic interneurons in the hippocampus. Additionally, we counted the inhibitory synapses labeled with vesicular inhibitory neurotransmitter transporter (VGAT) around the cell bodies of the principal neurons and found that L1 ameliorates the loss of inhibitory synaptic terminals.

## 2. Materials and Methods

### 2.1. Mice

We obtained three-month-old transgenic male APP/PS1 mice (C57BL/6J-TgN; Thy1-APPKM670/671NL; Thy1-PS1L166P) and their wild-type littermates from a breeding colony at the University of Tübingen, Germany. APP/PS1 mice co-express human APP containing a double Swedish (KM670/671NL) mutation, combined with the human PS-1-carrying L166P mutation [12]. These transgenes are expressed under the neuron-specific murine Thy-1 promoter, which is active postnatally. The mice were kept at the central animal facility of the University Hospital Hamburg-Eppendorf, under standard conditions. All experiments were conducted in accordance with the “Principles of laboratory animal care” (NIH publication No. 86-23, revised 1985), as well as with German and European Community laws on the protection of experimental animals. The experiments were approved by the responsible committee of the State of Hamburg (permit number 09/098) and by the Ethics Committee of the University of Kragujevac (permit number 01-617/4) and were conducted in association with our supervising veterinarian Dr. Andreas Haemisch. The overall number of mice used in the experiments was 10 APP/PS1 transgenic and 5 wild-type control mice, as groups of 5 animals were shown to be sufficient for statistical analysis in our previous experiments [5]. All experiments, data acquisition, and analyses were performed by a single experimenter, who was blinded with respect to genotype and treatment.

### 2.2. Viral Vectors

Viral vectors were provided by Dr. Sebastian Kügler, University of Göttingen, Germany. We used AAV5 vectors that were created to express L1 or GFP, as previously described [4]. We have previously determined that AAV-5 serotype has high diffusion and transduction rates [14]. The constructs were expressed under the murine cytomegalovirus (mCMV) promoter. The genome particle transducing unit ratio ranged from 25:1 to 35:1. The vectors were injected at a concentration of 3 × 109 transducing units/µL.

### 2.3. Surgery

Stereotactic injections of viruses were performed in a similar manner to that described previously [5]. Briefly, 3-month-old mice were anesthetized by an intraperitoneal injection containing a mixture of ketamine (100 mg/kg) and xylazine (5 mg/kg), placed in a stereotaxic frame (Stoelting, Dublin, Ireland), and the skull was exposed. Using a dental drill, a hole in the skull was drilled on the right side. Using a Hamilton syringe, AAV-L1 or AAV-GFP (1 µL solution/injection at a concentration of 3 × 10^9^ transducing units/µL) were injected unilaterally into the right hippocampus (Figure 1A). Working according to the Mouse Brain Atlas [21], we injected at the coordinates: −2 mm from Bregma, 1.5 mm from the midline, 1.5 mm deep. After surgery, the mice were kept on a hot plate (37 °C) for several hours to prevent hypothermia and were thereafter kept for 12 months, single-housed in a temperature-controlled room, with water and food ad libitum.

### 2.4. Tissue Fixation, Sectioning, and Immunostaining

Mice were terminally anesthetized with an intraperitoneal injection of a 16% solution of sodium pentobarbital (Narcoren, Merial, Hallbergmoos, Germany), 5 µL per 1 g body weight. After full terminal anesthesia was achieved, typically within 1 min of the injection, mice were perfused through the left heart ventricle with a fixative containing 4% formaldehyde solution in 0.1 M phosphate buffer pH 7.3 for 15 min at room temperature. The brains were then post-fixed in situ for 1 h, then removed afterward and left in the same fixative overnight. The following day, the brains were immersed in a 15% sucrose solution for 2 days. Afterward, they were immerse-frozen for 2 min in 2-methyl-butane precooled to −70 °C and stored at −80 °C until they were sectioned. The brains were cut into spaced-serial coronal 25-µm-thick sections on a cryostat (Leica CM3050, Leica Instruments, Nußloch, Germany), collected on SuperFrost Plus glass slides (Roth), and stored at −20 °C until staining. Immunofluorescent staining was performed as described previously [14]. Briefly, antigen retrieval (de-crosslinking of formaldehyde-fixed proteins) was performed in a water bath with 0.01 M sodium citrate water solution at pH 9.0 for 30 min, at 70 °C. Non-specific binding was then blocked in a solution of 0.2% triton X and 5% normal goat serum in 0.1 M phosphate-buffered saline at pH 7.3 (PBS) for a minimum of 1 h, at room temperature. Incubation with the primary antibody, diluted as indicated above, in the blocking solution was followed by 2 days at 4 °C. For the immunofluorescence experiments, we used the following primary antibodies: monoclonal mouse anti-NeuN (1:1000, Chemicon, Darmstadt, Germany, MAB377); monoclonal mouse anti-parvalbumin (1:1000; Sigma-Aldrich, Darmstadt, Germany, MAB1572); polyclonal rabbit anti-vesicular inhibitory transmitter transporter (VGAT) antibody (1: 1000; Synaptic Systems, Goetingen, Germany, 131002); and L1 555 monoclonal mouse antibody (1:200, a gift from Melitta Schachner). After washing for 3 × 15 min in PBS, the sections were incubated in appropriate fluorescently (Cy2- or Cy3-) tagged secondary antibody (goat anti-mouse or goat-anti rabbit), diluted 1:200 in PBS. After a subsequent wash in PBS (3 × 15 min), the sections were incubated for 10 min at room temperature in bis-benzimide solution (Hoechst 33258, Sigma-Aldrich) to counterstain for cell nuclei. After one final rinse in PBS, the sections were mounted in Fluoromount G (Southern Biotechnology Associates, Birmingham, AL, USA) and stored in the refrigerator until use. To control for the non-specific binding of the secondary antibody instead of the primary antibody, we incubated sections in a pre-immune serum from the animal in which the primary antibody was produced. These negative controls resulted in a complete absence of fluorescent signal.

### 2.5. Stereological Cell Counts and Analysis of Synapses

To estimate the cell densities (number of cells per unit volume) of NeuN- and parvalbumin-positive cells, we used the optical disector principle, as described previously [22]. All counts were performed using an Axioskop microscope (Carl Zeiss, Oberkochen, Germany) with a motorized stage, using a Stereo Investigator software-controlled computer system (Microbrightfield, Williston, VT, USA). To delineate the hippocampal subfields and layers, we observed nuclear staining (bis-benzimide) under low-power magnification. Every 10th section of the hippocampus was analyzed; overall, 4 sections were examined per animal, taken bilaterally (representing the whole dorsal hippocampus). In our serial spaced sections, every 10th section was 250 μm apart from the next; therefore, our analysis included representative sections for the 1-mm-thick dorsal hippocampal region of each animal. To estimate cell density, we counted the nuclei of the labeled cells found within systematically spaced optical disectors. The parameters for the analysis of parvalbumin-expressing cells were: guard space depth of 2 μm, base of the dissector 3600 μm^2^, and height of the disector 10 μm. The distance between the disectors was 60 μm. Similar parameters were used for the estimation of the density of NeuN-positive cells, apart from the base of the dissector, which was 900 μm^2^, and the space between dissectors, which was 90 μm, due to the very high density of these cells. After measuring the surface areas directly under the microscope, the volumes were calculated using Cavalieri’s principle. Since no differences between the volume of the hippocampus and the thickness of the hippocampal principal cell layers were found between the experimental and control groups, the cell densities that we show represent total cell numbers. Quantification of the perisomatic inhibitory synaptic terminals and the area of the principal cell bodies was performed as described previously [5,22]. Briefly, stacks of 1-μm-thick images were obtained from principal cell layers in different hippocampal subfields in VGAT/PV double immunofluorescently stained sections, using an LSM 510 confocal microscope (Zeiss). The images were standardized in size, taken using a 63 × 2 oil immersion objective at a 1024 × 1024-pixel resolution. One image per cell was used to measure the soma area and perimeter and to count the perisomatic puncta at the level estimated to be the largest cell body area. Numbers of VGAT-positive terminals were then normalized to the cell perimeter length. We counted at least 20 cells per hippocampus subfield per animal. The perimeter and area measurements were performed using ImageJ/Fiji 2.1 software (NIH).

### 2.6. Microscopy and Statistical Analysis

Images were taken on an Axiophot 2 microscope with a digital camera and Zen 2.3 software (Zeiss) or on an LSM 510 confocal microscope (Zeiss). The images were processed for brightness and contrast and cropped using Adobe Photoshop CS5 software (Adobe Systems Inc., San Jose, CA, USA). No further processing of the images was performed. All numerical data in this manuscript are shown as group mean values with standard deviation (SD). Data were first tested with the Shapiro–Wilk test to confirm normal distribution. If this was the case, statistical differences were determined by a one- or two-way analysis of variance (ANOVA), following the appropriate post hoc test if the difference was significant, using SYSTAT 9 software (SPSS 29). A one-way ANOVA was chosen because it effectively compares the means of multiple groups. A two-way ANOVA was used for the analysis of synaptic terminals, with the independent variable “parvalbumin expression” and the dependent variable “treatment”. Differences with a threshold lower than 5% were accepted as significant.

## 3. Results

### 3.1. AAV-L1 Injection Reduces Aβ Plaque Load in the Hippocampus of APP/PS1 Mice

Three-month-old transgenic male APP/PS1 mice were injected with AAV-L1 or with AAV-GFP as a control. After sacrifice at 15 months, we fixed the brains and then used them for histological analysis (Figure 1A). Three months after the AAV-GFP injection, we could identify GFP expression in the hippocampus with the strong transduction of neurons (Figure 1B,C). Importantly, we also stained AAV-L1-injected hippocampus sections with the antibody against L1. While, in the wild-type mouse, L1 expression in the CA1 subfield of the hippocampus was confined to the interneurons (Figure 1D), the AAV-L1 transduced hippocampus showed robust L1 expression in the pyramidal cells (Figure 1E). We also stained the APP/PS1 mouse brain sections for amyloid plaques and could show that, as in our previous study [5], the AAV-L1 injected mice had fewer plaques and a smaller overall plaque-covered area compared to the AAV-GFP-injected controls.

### 3.2. Aav-L1 Injection Reduces the Loss of NeuN+ Granule Cells in the Dentate Gyrus of App/Ps1 Mice

The primary aim of our study was to compare the AAV-L1 treatment and the control AAV-GFP samples from the APP/PS1 mice. However, since in these mice, the neuronal loss could occur at different rates, we compared APP/PS1 mice with a control group of wild-type mice. As an additional control, in all experiments, we made quantifications on the side contralateral to the injection site in both AAV-GFP- and AAV-L1-injected groups, which did not statistically differ from the AAV-GFP injected groups. Another control was an injection of saline solution into the wild-type hippocampus, which produced similar results as for the wild-type group (Appendix A). Our results indicate that there was no loss of CA1 pyramidal cells (one-way ANOVA; F = 1.28, *p* = 0.315) and CA3 pyramidal cells (one-way ANOVA; F = 1.351, *p* = 0.293) in the APP/PS1 mice compared with control and, thus, no difference between the AAV-L1 and AAV-GFP-injected groups (Figure 2A–D). In the dentate gyrus, however, the number of granule cells was reduced in the AAV-GFP-injected mice, but not in the AAV-L1-injected mice (Figure 2E,F; one-way ANOVA; F = 9.795, *p* < 0.001; post hoc WT vs. AAV-L1, *p* = 0.116; WT vs. AAV-GFP, *p* = 0.003; AAV-L1 vs. AAV-GFP, *p* = 0.045). Thus, in the granule cells of the dentate gyrus, the number of cells in the AAV-L1 injection group was significantly higher than in the AAV-GFP-injected control group (*p* < 0.05). We conclude that, although there is no massive generalized neuronal loss in the hippocampus, as is congruent with previous findings [10], AAV-L1 injections protected APP/PS1 mice from granule cell loss.

### 3.3. AAV-L1 Injection Reduces Parvalbumin + Interneuron Loss in the Hippocampus of App/Ps1 Mice

As in many neurological disease models, in APP/PS1 mice, the inhibitory interneurons of the hippocampus are affected before general neuronal loss occurs [11]. As parvalbumin-expressing interneurons represent a very important subpopulation of hippocampal interneurons (amounting to about 40%), are functionally very important and are the most vulnerable in various neurological disorders [22,23], we examined this subpopulation of inhibitory interneurons in AAV-L1 and AAV-GFP-treated mice, as well as wild-type control mice. In the AAV-GFP-injected APP/PS1 mice, the number of PV+ interneurons was lower than in the controls in all 3 subfields (Figure 3). Congruently, in all 3 subfields, AAV-L1 injection led to a better outcome (more PV+ cells) compared to AAV-GFP in APP/PS1 mice (Figure 3B,D,F). In the CA1 subfield (one-way ANOVA; F = 8.183, *p* = 0.002; post hoc WT vs. AAV-L1 *p* = 0.63; WT vs. AAV-GFP *p* = 0.005; AAV-L1 vs. AAV-GFP *p* = 0.011) and DG subfield (one-way ANOVA; F = 6.439, *p* = 0.005; post hoc WT vs. AAV-L1 *p* = 0.72; WT vs. AAV-GFP *p* = 0.014; AAV-L1 vs. AAV-GFP *p* = 0.042), the number of PV+ neurons in the AAV-L1-injected APP/PS1 mice was not significantly different from the wild-type controls (Figure 3C,F), whereas, in the CA3 subfield (one-way ANOVA; F = 13.741, *p* < 0.001; post hoc WT vs. AAV-L1 *p* = 0.045; WT vs. AAV-GFP *p* < 0.001; AAV-L1 vs. AAV-GFP *p* = 0.021), the number was still lower than in the wild-type mice (Figure 3D). Therefore, in the dentate gyrus, CA1 and CA3, the number of cells in the AAV-L1 injection group was significantly higher than in the control group (*p* < 0.05). We conclude that PV + interneuron loss in the APP/PS1 mice can be alleviated by AAV-L1 treatment.

### 3.4. L1-AAV Injection Leads to Higher Numbers of Inhibitory Synapses Around Pyramidal Cells in the CA1 and CA3 and Granule Cells in the DG Subfield in APP/PS1 Mice

As we observed a significant loss of inhibitory interneurons in the APP/PS1 mice, we next estimated the numbers of inhibitory synapses around the cell bodies of principal neurons in the CA1, CA3, and DG hippocampus subfields. Therefore, we immunostained the sections for the vesicular inhibitory transmitter transporter, VGAT, which labels all inhibitory presynaptic terminals, both GABAergic and glycinergic, in the hippocampus [5]. Additionally, since one of the most important sources of inhibition of principal cells in the hippocampus is parvalbumin-expressing interneurons [22], we performed double immunofluorescence staining with VGAT and parvalbumin antibodies. In that way, we were able to demonstrate parvalbumin-negative (single-stained for VGAT) and parvalbumin-positive (double-stained for VGAT and PV) axonal terminals (Figure 4A,C,E). When compared to the control wild-type mice, the number of both PV+ and PV- terminals around pyramidal cells in the CA1, CA3, and DG subfields of the hippocampus was significantly reduced in APP/PS1 mice (Figure 4B,D,F). Likewise, the numbers of PV+ and PV- terminals were higher in AAV-L1 injected animals than in AAV-GFP injected controls (Figure 4B,D,F). The numbers of VGAT+ inhibitory synapses were thus still significantly lower in the CA1 and DG of AAV-L1 injected mice compared to wild-type controls (Figure 4B,F), whereas in the CA3 subfield, this difference was not significant (Figure 4D). A two-way ANOVA with the factors “treatment” and “parvalbumin expression” for the CA1 subfield has shown significance for both factors (*p* < 0.001) and no significance for their interaction (*p* = 0.247) and post hoc was significant for all pairwise comparisons (*p* < 0.001). The CA3 two-way ANOVA also detected significance for both factors (*p* < 0.001) and no significance for their interaction (*p* = 0.169) and post hoc was significant for all pairwise comparisons (*p* < 0.001) except for AAV-L1 vs. WT within the PV+ group (*p* = 0.175). The DG two-way ANOVA detected significance for both factors (*p* < 0.001) and their interaction (*p* < 0.001) and post hoc was significant for all pairwise comparisons (*p* < 0.001). Thus, the number of inhibitory synaptic terminals around pyramidal cells in the CA1 and CA3 regions and around the granule cells of the DG was significantly higher in the AAV-L1 injection group than in the control group (*p* < 0.001). We conclude that AAV-L1 injection significantly improves the survival/rearrangement of inhibitory synapses in the hippocampus of APP/PS1 mice, which has potential functional implications.

## 4. Discussion

In this study, we show that the neural adhesion molecule L1 promotes neuronal survival, which is particularly true for inhibitory interneurons, as well as the preservation and/or remodeling of inhibitory synapses in a mouse model of Alzheimer’s disease. This study is an extension of our previous work, in which we followed AAV-L1 mice injected at 3 months, as here, until they were 8 months old [5]. In that study, L1 was found to be overexpressed in neurons and ectopically expressed in astrocytes [5]. Histological analysis revealed a decreased Aβ plaque load in AAV-L1-injected mice, as well as reduced astrogliosis and increased densities of inhibitory synaptic terminals on pyramidal cells in the hippocampus when compared with AAV-GFP injected controls. This was most likely due to the direct binding and scavenging of Aβ peptides [5]. We now extend the observation of decreased Aβ plaque load in AAV-L1-injected mice to a time point of 15 months, 12 months after the injection. Additionally, at this time point, we could study the effect of L1 on neuron and synaptic loss in APP/PS1 mice.

In the APP/PS1 mice, there was no cortical neuron loss up to 8 months of age, but local neuron loss in the dentate gyrus was demonstrated [10,12]. Despite significant amyloidosis in the hippocampus, they displayed a rather subtle behavioral phenotype, i.e., impairment in working memory at 8 months of age [12]. Congruently, they showed impaired long-term potentiation in the CA1 in vivo [13]. The principal neuron loss in this mouse model has been limited to the dentate gyrus [10]. Our results confirm this finding and show that L1 viral transduction rescues this effect. In a similar APP/PS1 double transgenic model, the loss of parvalbumin-positive interneurons was reported in the olfactory cortex at this age [24], and physiological anomalies in the parvalbumin-positive cortical interneurons were recorded [25]. Selective vulnerability of parvalbumin-positive interneurons is known to occur under different experimental conditions, such as epilepsy [26], interleukine 6 overexpression [27], and the ketamine schizophrenia model [27]. In this study, we recorded a loss of parvalbumin-positive interneurons in the hippocampus, which was not observed at 8 months of age [5]. Importantly, L1 seems to be instrumental in preserving these interneurons. This effect could be of functional significance as parvalbumin-expressing interneurons are critically important for excitation/inhibition balance and functional working memory in the hippocampus [28,29]. Hippocampal interneurons play a crucial role in regulating the excitability of pyramidal neurons, which are essential for memory formation and spatial navigation [30]. As parvalbumin-expressing interneurons are particularly important for long-term potentiation, an electrophysiological mechanism of learning and memory [22,29], we can speculate that better preservation of this cell population would indeed lead to better performance in behavioral tests.

In our study, we show the differences between hippocampus subfields in principal neuron loss, which seems to affect the dentate gyrus before the rest of the hippocampus in APP/PS1 mice. It has been shown that in AD patients and in various mouse AD models, the dentate gyrus shows early neurodegeneration, even in the relative absence of pathological hallmarks (amyloid plaques and tau tangles) of the disease [30]. Furthermore, APP/PS1 mice show early electrophysiological changes, i.e., the reduced plasticity of intrinsic (non-synaptic) excitability in the dentate gyrus [31]. Reduction of the inhibition (numbers of parvalbumin-expressing interneurons and inhibitory synapses) in all hippocampus subfields that we detected in our study is, therefore, a change that precedes principal neuron and excitatory synaptic loss.

Because the early signs of AD pathology occur at a synaptic level, the loss of dendritic spines and synaptic terminals is a commonly seen pathology in AD mouse models [26]. Therefore, we were not surprised to find significantly reduced numbers of inhibitory perisomatic terminals in all hippocampal subfields, compared with the wild-type controls. Interestingly, this loss was not observed in the dentate gyrus. Notably, AAV-mediated L1 overexpression reduced the loss of VGAT-expressing presynaptic terminals in all hippocampus subfields, which, considering the functional significance, is the most remarkable effect of L1 overexpression in APP/PS1 mice. This beneficial effect of L1 is consistent with the previously reported effects of L1 as a trophic factor for neurons and a promoter of neurite outgrowth during development and upon injury, as well as an enhancer of synaptic plasticity, both in vitro and in vivo [5,7,15]. This is congruent with the finding that, although APP/PS1 mice suffer the significant loss of interneurons, the loss of principal cells in the hippocampus is relatively moderate and is limited to the dentate gyrus [10] (this study).

Our choice of AAV-mediated gene transfer for the proof-of-concept that L1 mitigates neurodegeneration in an AD model in our study was guided by the fact that it is easily applicable to deep brain structures like the hippocampus. It is a matter of ongoing debate if such an approach could be clinically relevant [32]. Besides this and our previous study [4], there have been several other works utilizing the AAV-mediated gene transfer approach to various AD models. For example, in the same mouse model that we used, the AAV transduction of Gas6 alleviated the amyloid plaque load [33]. Similarly, nerve growth factor AAV transduction to the hippocampus prevented cholinergic neuron loss and improved behavioral performance in rats with cholinergic deficit [34]. This demonstrates that, given an early diagnosis, AAV could be a potent therapeutic tool for neurodegenerative disorders, including AD [32].

PET imaging of the fibrillar amyloid-beta allows for the detection of fibrillar Aβ deposition in vivo and enables the early clinical or even preclinical detection of the disease and the accurate distinction of AD from other dementias [35,36,37]. As in those patients, the loss of neurons and their circuitries has yet to occur; early diagnosis gives hope for more successful future treatment strategies. Early treatment could postpone or slow down the disease progression and development of the pathological hallmarks of the disease. From the viewpoint of amyloid clearance enhancement, or amyloid production suppression, some key questions remain unanswered. Importantly, it is not clear as to what extent the lowering of Aβ peptide content in the brain should be able to mediate a disease-modifying effect. Additionally, it remains unknown at what stage of the disease an amyloid-β-directed treatment approach would show therapeutic effects. It is, however, clear that the L1 protein or L1-function-mimicking compounds, such as the recently published L1-mimicking peptide [38], also have, in addition to the Aβ clearance effect, direct neuroprotective effects; therefore, they are well-suited as potential candidates for therapy in AD. It is noteworthy in this context that the newly discovered roles of the proteolytic cleavage of L1 could further enable the use of smaller L1-based components, which could be easier to apply [38,39,40].

## 5. Conclusions

To conclude, we have evaluated the therapeutic potential of L1 in a mouse model of cerebral amyloidosis. L1 has already been shown to bind with Aβ peptides and reduce plaque load, astrogliosis, and the Aβ-42/Aβ-40 ratio in 8-month-old APP/PS1 mice after being injected at 3 months of age [5]. We have now increased the survival period to 15 months, in order to wait for the mice to develop more severe neurodegeneration. We have shown that L1 protects the neurons, including dentate gyrus granule cells and parvalbumin-expressing interneurons, as well as inhibitory synapses in the hippocampus. Our manuscript is based on the analysis of structural changes in an AD model mouse, and we can only speculate that morphological improvement upon AAV-L1 treatment would be followed by a favorable behavioral outcome. This is plausible, as APP/PS1 mice show an early and relatively robust behavioral phenotype [12,13], and we believe that our structural analysis of the synaptic terminals underlying physiological and behavioral changes is suggestive of a better behavioral outcome in L1-treated mice. Another limitation of our study was the lack of comparison with other potentially beneficial treatments in AD models, particularly because we do not propose that L1 could be used as a single therapy, but rather in combination with other agents. In addition, the use of the whole L1 protein through AAV-mediated gene transfer could be impractical in clinical settings. Our findings, however, add to the evidence showing that, providing a suitable way of application is developed, L1 could be used in Alzheimer’s disease treatment, due to its neuroprotective and amyloid peptide-binding properties.

## Figures and Tables

**Figure 1 biomedicines-12-01726-f001:**
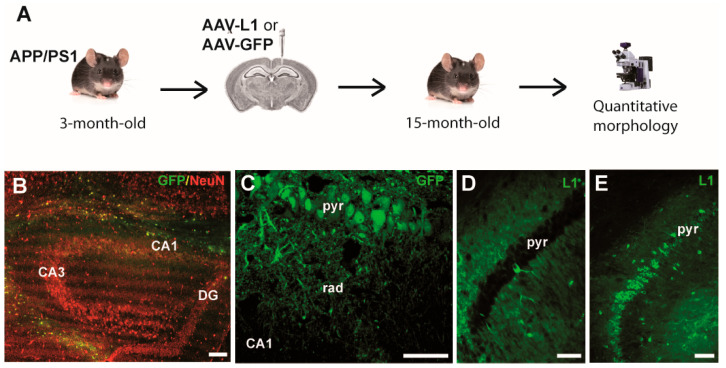
(**A**) Schematic depiction of the experimental design. (**B**) Representative image of the APP/PS1 mouse hippocampus injected with AAV-GFP (green) and immunofluorescently stained for NeuN (red). The overlay (yellow) highlights the transduced cells. (**C**) Higher magnification of the CA1 hippocampal subfield, with GFP (green) transduced pyramidal cells. (**D**,**E**). Immunostaining with L1 555 antibody (green) in wild-type (**D**) and AAV-L1 injected (**E**) hippocampus cells. Note the robust transduction of the CA1 pyramidal cells with AAV-L1. CA—cornu ammonis, DG—dentate gyrus, pyr—stratum pyramidale, rad—stratum radiatum. Scale bars: 200 µm (**B**); 25 µm (**C**–**E**).

**Figure 2 biomedicines-12-01726-f002:**
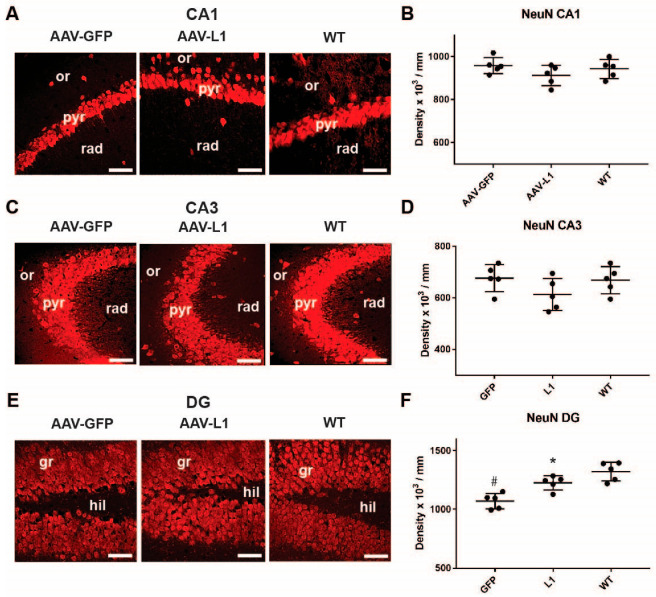
Injection of AAV-L1 reduces the loss of hippocampal NeuN-positive granule neurons in the dentate gyrus of APP/PS1 mice. (**A**,**C**,**E**) Representative images of NeuN-immunostained (NeuN+) neurons in the CA1 (**A**), CA3 (**C**), and DG (**E**) subfields of the hippocampus. Or—stratum oriens, pyr—stratum pyramidale, rad—stratum radiatum, gr—stratum granulosum, hil—hilus of the dentate gyrus. Scale bar: 50 µm. (**B**,**D**,**F**) Densities of the hippocampal NeuN-positive neurons in the pyramidal layer of the CA1 (**B**), CA3 (**D**), and granule cells in the DG (**F**) in wild-type (WT), AAV-L1, and AAV-GFP-injected APP/PS1 mice. Data are shown as mean + standard deviation. Asterisks indicate the differences between treatments, while the hashtag indicates a difference from the wild-type control; one-way ANOVA with Holm–Sidak post hoc, *p* < 0.05; n = 5 mice/group.

**Figure 3 biomedicines-12-01726-f003:**
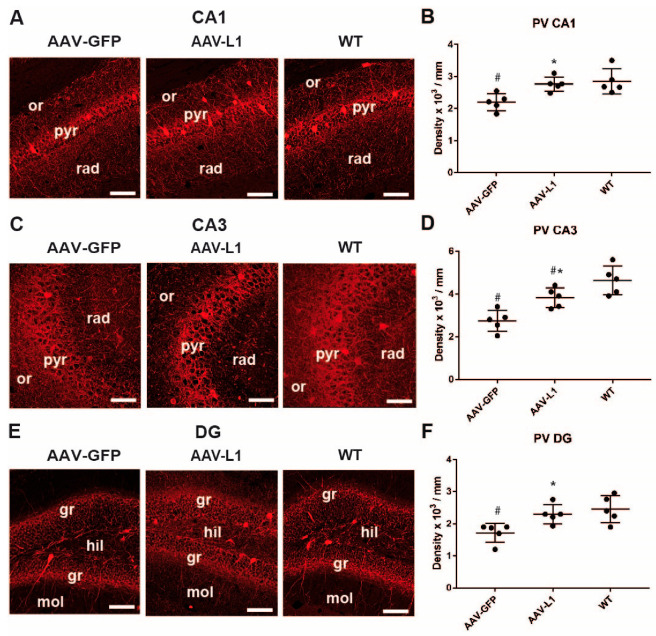
Injection of AAV-L1 reduces the loss of hippocampal parvalbumin-positive interneurons in APP/PS1 mice. (**A**,**C**,**E**) Representative images of parvalbumin-immunostained (PV+) interneurons in the CA1 (**A**), CA3 (**C**), and DG (**E**) subregions of the hippocampus. Or—stratum oriens, pyr—stratum pyramidale, rad—stratum radiatum, gr—stratum granulosum, mol—stratum moleculare, hil—hilus of the dentate gyrus. Scale bar: 25 µm. (**B**,**D**,**F**) Densities of PV+ neurons in the CA1 (**B**), CA3 (**D**), and DG (**F**) in the wild-type (WT) and AAV-L1 or AAV-GFP-injected APP/PS1 mice. Data are shown as mean + standard deviation. Asterisks indicate the difference between treatments, the hashtag indicates a difference from the wild-type control; one-way ANOVA with Holm–Sidak post hoc, *p* < 0.05; n = 5 mice/group.

**Figure 4 biomedicines-12-01726-f004:**
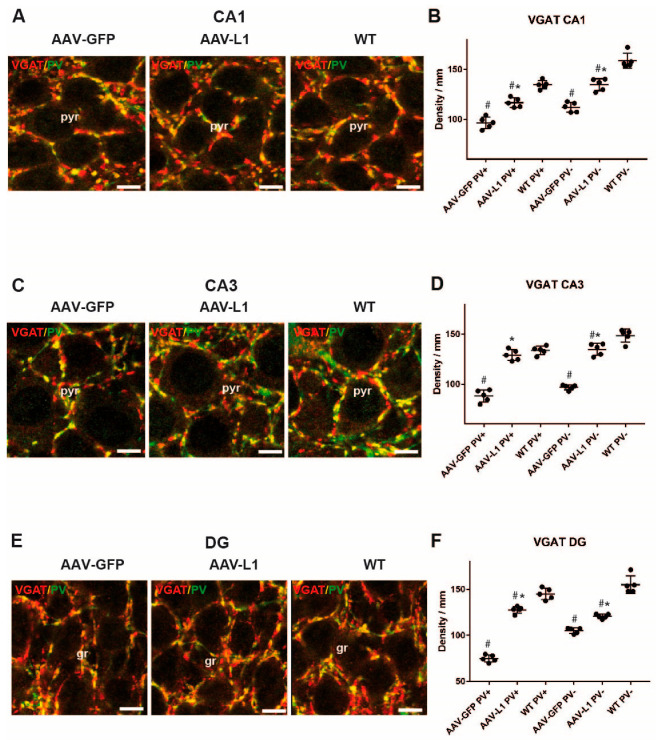
The loss of perisomatic inhibitory synapses on principal neuron cell bodies in the hippocampus of APP/PS1 mice is reduced by AAV-L1 injection. (**A**,**C**,**E**) Representative confocal micrographs of VGAT-(red) and parvalbumin (PV, green)-immunostained perisomatic terminals around CA1 (**A**), CA3 (**C**), pyramidal neurons (pyr) and DG granule cells (gr) (**E**). Scale bar: 10 µm. (**B**,**D**,**F**) Diagrams represent the number of parvalbumin-positive/VGAT-positive (PV+) and parvalbumin-negative/VGAT-positive (PV-) perisomatic terminals per unit length (mm) in the CA1 (**B**), CA3 (**D**), and DG (**F**) of wild-type (WT) mice, and APP/PS1 mice injected with either AAV-L1 or AAV-GFP. Data are shown as mean + SD. Asterisks indicate a difference between the injections (AAV-L1 or AAV-GFP), hashtags indicate a difference between APP/PS1 mice and the wild-type control, in a two-way ANOVA with the factors “parvalbumin expression” and “viral injection”, followed by the Holm–Sidak post hoc method, *p* < 0.05; n = 5 mice/group.

## Data Availability

Original data are available from the corresponding authors upon request.

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
