# Peer review of "The Neuroprotective Effect of Neural Cell Adhesion Molecule L1 in the Hippocampus of Aged Alzheimer’s Disease Model Mice"

_biomedicines, 2024, doi:10.3390/biomedicines12081726_

Round 1

Reviewer 1 Report

Comments and Suggestions for Authors

The manuscript titled "The neuroprotective effect of neural cell adhesion molecule L1 in the hippocampus of aged Alzheimer’s disease model mice" investigates the effects of L1 on amyloid plaque load, neuronal survival, and synaptic preservation in a transgenic mouse model of Alzheimer's disease. The study is well-structured and provides valuable insights into potential therapeutic approaches for Alzheimer's disease. Based on the review of the provided manuscript, here are my comments and suggestions:

1. Abstract: When summarizing the main findings, include a statement such as "Our results show that AAV-L1 injection significantly promoted the survival of inhibitory synapses in all three hippocampal subregions compared to the AAV-GFP control group (p < 0.05)."

2. Introduction: Enhance the literature review by incorporating more recent studies on the therapeutic potential of neural adhesion molecules in neurodegenerative diseases.

3. Methods: In the statistical analysis section, add an explanation such as "One-way ANOVA was chosen because it effectively compares the means of multiple groups."

4. Results: For instance, when describing specific findings, state "In the granule cells of the dentate gyrus, the number of cells in the AAV-L1 injection group was significantly higher than in the control group (p < 0.01)."

5. Consistency in Abbreviations: Create a list of abbreviations and define them upon first use, maintaining consistency throughout the text.

Author Response

We thank the Reviewer for their comments and suggestions to improve our manuscript. Point-to-point responses:

Reviewer 1

The manuscript titled "The neuroprotective effect of neural cell adhesion molecule L1 in the hippocampus of aged Alzheimer’s disease model mice" investigates the effects of L1 on amyloid plaque load, neuronal survival, and synaptic preservation in a transgenic mouse model of Alzheimer's disease. The study is well-structured and provides valuable insights into potential therapeutic approaches for Alzheimer's disease. Based on the review of the provided manuscript, here are my comments and suggestions:

  1. Abstract: When summarizing the main findings, include a statement such as "Our results show that AAV-L1 injection significantly promoted the survival of inhibitory synapses in all three hippocampal subregions compared to the AAV-GFP control group (p < 0.05)."

R: We made the corrections as the Reviewer suggested.

  1. Introduction: Enhance the literature review by incorporating more recent studies on the therapeutic potential of neural adhesion molecules in neurodegenerative diseases.

R: We added some more recent references, as the Reviewer suggested (ln 71-76; ln 105-107).

  1. Methods: In the statistical analysis section, add an explanation such as "One-way ANOVA was chosen because it effectively compares the means of multiple groups."

R: We revised the section accordingly (ln 239-242).

  1. Results: For instance, when describing specific findings, state "In the granule cells of the dentate gyrus, the number of cells in the AAV-L1 injection group was significantly higher than in the control group (p < 0.01)."

R: We revised the Results accordingly.

  1. Consistency in Abbreviations: Create a list of abbreviations and define them upon first use, maintaining consistency throughout the text.

R: We made a list of abbreviations, and we submitted it as supplementary material. If the journal regulations allow, we would publish it as a footnote. We apologize for the previous inconsistencies.

Reviewer 2 Report

Comments and Suggestions for Authors

The authors presented “The neuroprotective effect of neural cell adhesion molecule L1 in the hippocampus of an aged Alzheimer’s disease model." Overall, the findings are interesting. There are several major comments on experimental design that need to be addressed before considering it.

Strengths of the study: The authors have shown the therapeutic potential of cell adhesion molecule L1 using an AD mouse model.

Limitations of the study: The author didn’t use a positive control or standard drug to compare L1 therapeutic potential.

An additional set of experiment is required.

Abstract:

Authors must add the route of drug administration.

Introduction:

The authors must give a little bit of epidemiology and available treatments, along with recent ongoing research and discoveries.

The author must elaborate on why they have chosen L1 for therapeutic purposes with other evidence.

Include more about how they measured a vesicular inhibitory transmitter like VGAT.

Methodology:

The author must include a saline-injected control group to observe the effects of stereotactic surgery.

They must include the treatment of the standard drug group to compare the therapeutic potential of L1.

Why did they take only 5 mice in the WT group, which makes the study hard to conclude?

How could authors conclude their results based on 10 AD and 5 WT mice: Please explain.

What was the dose of ketamine and xylazine?

Why are there different anesthetics (sodium pentobarbital)? used for tissue fixation? There must be a differential effect on outcome. How did the authors control that?

Results:

3.1 Please correct the heading; beta is missing there.

Arrange the figures as per the results sequence.

It seems there is an increase in hil cells in the DG region of AAV-GFP (Figure 2e), which contradicts Figure 2f. please explain?

Why was Holm-Sidak posthoc used and not Tukey’s posthoc?

Discussion and conclusion:

Line 382: The whole paragraph must go to the introduction section and here the author must discuss why and how they used.

I would recommend that authors discuss their findings with the existing evidence.

Please include the limitations of the current study.

Author Response

We thank the Reviewer for their comments and suggestions to improve our manuscript. Point-to-point responses:

Reviewer 2

The authors presented “The neuroprotective effect of neural cell adhesion molecule L1 in the hippocampus of an aged Alzheimer’s disease model." Overall, the findings are interesting. There are several major comments on experimental design that need to be addressed before considering it.

Strengths of the study: The authors have shown the therapeutic potential of cell adhesion molecule L1 using an AD mouse model.

Limitations of the study: The author didn’t use a positive control or standard drug to compare L1 therapeutic potential.

An additional set of experiment is required.

Abstract:

Authors must add the route of drug administration.

R: We added that the viruses were administered by intracerebral injection.

Introduction:

The authors must give a little bit of epidemiology and available treatments, along with recent ongoing research and discoveries.

The author must elaborate on why they have chosen L1 for therapeutic purposes with other evidence.

Include more about how they measured a vesicular inhibitory transmitter like VGAT.

R: We modified the Introduction according to the Reviewer’s suggestions (ln 50-52; ln 71-76; ln 105-107).

Methodology:

The author must include a saline-injected control group to observe the effects of stereotactic surgery.

R: We added, as supplementary figure 1 our data comparing the virus injection to saline injection, showing no statistical difference to non-injected wild-type mice.

They must include the treatment of the standard drug group to compare the therapeutic potential of L1.

R: We agree with the Reviewer that it would be optimal to compare our results with another accepted therapeutic agent. It is, however, very difficult, given our unique way of delivery to choose a therapeutic agent, given that there are few therapies that are approved in patients, and many more experimental studies with variable results. Furthermore, those experiments would have to be done in parallel in order to be completely comparable, which exceeds our current resources. We added a paragraph to the Discussion, comparing some more prominent experimental studies of comparable design to our results (ln 425-433).

Why did they take only 5 mice in the WT group, which makes the study hard to conclude?

How could authors conclude their results based on 10 AD and 5 WT mice: Please explain.

R: All three groups in our experiments had 5 mice (5 APP/PS1 mice injected with AAV-L1, 5 with AAV-GFP and 5 WT mice). In our previous experience, the size of the groups was sufficient to get significant results.

What was the dose of ketamine and xylazine?

R: We added the doses of both anesthetics. We apologize for the omission.

Why are there different anesthetics (sodium pentobarbital)? used for tissue fixation? There must be a differential effect on outcome. How did the authors control that?

R: The terminal anesthesia with pentobarbital before tissue fixation takes typically only a minute between the injection and unresponsiveness, upon which time is the animal sacrificed. This is too short time for morphological changes. Additionally, all animals in the study were treated in the same way. We added a short explanation of this to the Materials and Method section.

3.1 Please correct the heading; beta is missing there.

R: We added beta, we apologize for this omission.

Arrange the figures as per the results sequence.

R: We arranged the figures so that they now follow the sequence on the graphs.

It seems there is an increase in hil cells in the DG region of AAV-GFP (Figure 2e), which contradicts Figure 2f. please explain?

R: We checked again with our source files and noticed that during the revision of the previous submission of the manuscript the two panels got mixed up. We apologize and we have now corrected the error.

Why was Holm-Sidak posthoc used and not Tukey’s posthoc?

R: Our statistical program choses the most appropriate posthoc test based on the data.

Line 382: The whole paragraph must go to the introduction section and here the author must discuss why and how they used.

R: The paragraph was moved to the Introduction.

I would recommend that authors discuss their findings with the existing evidence.

 R: We broadened the Discussion by adding further relevant references (ln 423-433).

Please include the limitations of the current study.

R: We added the limitations of the current study to the concluding paragraph (ln 455-459).

Round 2

Reviewer 2 Report

Comments and Suggestions for Authors

I appreciate that the authors tried to address most of the comments. However, I would recommend presenting the data in an individual dot plot rather than a bar graph.

Further, I haven't seen any behavioral tests performed that confirm the cognitive deficits, which is important for model validation.

Although, author has used transgenic mice, an additional experiment is required to validate the model with a wild-type control.

Author Response

Comment 1: I appreciate that the authors tried to address most of the comments. However, I would recommend presenting the data in an individual dot plot rather than a bar graph.

Reply 1: We agree with the Reviewer and we now present our data as dot plot diagrams, as the Reviewer recommended.

Comment 2: Further, I haven't seen any behavioral tests performed that confirm the cognitive deficits, which is important for model validation. Although, author has used transgenic mice, an additional experiment is required to validate the model with a wild-type control.

Reply 2: Our manuscript deals with the structural phenotype (neurodegeneration) in APP/PS1 mice. We appreciate that it is a weakness of our manuscript that we performed no behavioral studies, for which we have neither funding nor expertise, and we now acknowledge the lack of behavioral experiments as a weakness of our manuscript in a conclusion paragraph (lines 453-459).

Round 3

Reviewer 2 Report

Comments and Suggestions for Authors

The authors have addressed all my comments.